Didymium arenosum, a myxomycete new to science from the confluence of deserts in northwestern China

Wei Shuwei 1 2
Li Shu 1 3
Liu Pu 1
Qi Bao 1
Wang Qi qiwang@jlau.edu.cn 1
Li Yu 1
1 Engineering Research Center of Chinese Ministry of Education for Edible and Medicinal Fungi, Jilin Agricultural University , Changchun , Jilin , China
2 College of Plant Protection, Jilin Agricultural University , Changchun , Jilin , China
3 Northeast Normal University , Changchun , Jilin , China
Puia Zothan
Electronic publication date: 2024 Jan 8
Publication date: 2024
Volume: 12
Electronic Location ID: e16725
Received 2023 May 25; Accepted 2023 Dec 5
Copyright: ©2024 Wei et al.
Copyright year: 2024
Copyright holder: Wei et al.
License: This is an open access article distributed under the terms of the Creative Commons Attribution License, which permits unrestricted use, distribution, reproduction and adaptation in any medium and for any purpose provided that it is properly attributed. For attribution, the original author(s), title, publication source (PeerJ) and either DOI or URL of the article must be cited.
License URL: https://creativecommons.org/licenses/by/4.0/

Keywords: Didymium, Morphology, Phylogeny, Scanning Electron Microscope, SSU rDNA, The life cycle

Funding: National Key R&D Program of China 2021YFD1600401 Science and Technology Development Project of Jilin Province 20190201197JC National Natural Science Foundation of China No. 31770011 This work was supported by the National Key R&D Program of China (2021YFD1600401); Science and Technology Development Project of Jilin Province (20190201197JC); and the National Natural Science Foundation of China (No. 31770011). The funders had no role in study design, data collection and analysis, decision to publish, or preparation of the manuscript.

==============================
A new myxomycete species, Didymium arenosum, was described based on morphological evidence and phylogenetic analyses. The species was discovered in the arid region at the confluence of the Badain Jaran desert and Tengger desert on the leaves of Betula platyphylla and was cultivated in a moist chamber culture. Morphologically, the species is distinguished by the greenish-yellow calcium carbonate crystals on the surface and the spores covered with small warts, some of which are connected into a short line. A phylogenetic analysis of D. arenosum strongly supports its classification as a separate clade. The spore to spore agar culture of D. arenosum requires 23 days, and this study provides a detailed description of its life cycle.

Introduction

Myxomycetes are a protist group belonging to the Eumycetozoa (Kang et al., 2017; Hehmeyer, 2019). They have played a crucial ecological role in terrestrial habitats and some of those metabolites have also been demonstrated to exhibit antitumor activity, antibacterial activity, and antioxidant activity (Keller & Everhart, 2010; Li et al., 2022; Wang, Li & Liu, 2022). Populations of these organisms are primarily regulated by food-related feeding practices, contributing to the maintenance of species diversity within their biosphere (Foissner & Hawksworth, 2009; Stephenson, Laursen & Seppelt, 2007). The investigation of slime molds in arid regions is mainly limited to certain areas in America, Asia, and Europe due to the difficulty of collections (Estrada-Torres et al., 2009; Lado et al., 2013; Lado et al., 2016; Novozhilov & Schnittler, 2008; Stephenson et al., 2020; Wrigley de Basanta, Lado & Estrada-Torres, 2010; Wrigley de Basanta, Lado & Estrada-Torres, 2012). Nevertheless, these studies have laid the foundation for exploring the diversity of myxomycetes in arid regions (Estrada-Torres et al., 2009; Wrigley de Basanta et al., 2009).

Didymium (Didymiaceae) was initially described in the 18th century, and to date, more than 90 species have been reported (Janik et al., 2021; Kirk et al., 2008; Lado, 2023). Previous studies indicated that Didymium is more likely to survive in arid areas than in other areas. For instance, nearly 20% of the species of Didymium were found in the arid and semi-arid regions of the Canary Islands. In addition, more than 20% of the species were found in the Twakan-Quikateland Valley, a biosphere reserve in Mexico (Estrada-Torres et al., 2009). However, recent studies have shown that most Didymium species are found in temperate and tropical climate regions (Wrigley de Basanta et al., 2015; Wrigley de Basanta, Lado & Estrada-Torres, 2008). China has a vast territory and abundant resources, but until now, only Schnittler has investigated the diversity of slime molds in the Tarim Basin of Xinjiang. In the future, it is necessary to explore further the species diversity of slime molds in the arid region of China, including the Gurbantunggut Desert, the Qaidam Basin Desert, and the Kubuqi Desert, etc (Schnittler et al., 2013; Zhang et al., 2020).

Minqin County is situated on the confluence of Badain Jaran desert and Tengger desert, which is an arid area in northwestern China, at 38°03′–39°28′N and 101°49′–104°12′E (Sun et al., 2005). The region undergoes an arid continental climate (Ma et al., 2007) with an annual average precipitation of less than 150 mm and annual evaporation exceeding 2,500 mm (Sun et al., 2007). Evaporation occurs at a rate approximately 20 times greater than precipitation. The elevation ranges from 1,280 to 1,477 m, and the average annual temperature is 8.3 °C (Xue et al., 2015). Vegetation coverage in arid regions is less than 10%, dominated primarily by Artemisia arenaria DC., Ephedra prezewalskii Stapf, Haloxylon ammodendron (C.A. Mey.) Bunge, and Nitraria tangutorum Bobr. (Chang et al., 2006). These plants are important for the protection and management of ecosystems in semiarid and arid regions (Wu et al., 2021).

Myxomycetes are rare in arid regions (Schnittler et al., 2013). However, recent research showed that 39 species of slime molds were discovered in arid areas of northwestern China from 2016 to 2018, of which only four belonged to the Didymium (Wei et al., 2019). The collection of these myxomycetes were obtained from live tree bark, deciduous, and dead branches, then through moist chamber culture using Cavender’s methods (Cavender, 1995). These findings provide valuable insights into the distribution and diversity of myxomycetes in arid regions, and our research will contribute to the understanding of these fascinating organisms.

Slime molds’ life cycles undergo transformations, progressing from trophic, diploid multinuclear plasmodia to reproductive, haploid fruiting bodies (Keller, Everhart & Kilgore, 2022). It has the capability to directly develop into haploid plasmodium, or it can generate diploid zygotes through heterothallism or homothallism. These then gradually evolve into amorphous, multinuclear plasmodia through continuous mitosis or fusion with other myxamoebae (Keller, Everhart & Kilgore, 2022). However, the detailed morphogenesis of only a few species has been described so far primarily because plasmodium retrieval from some species is difficult, especially regarding the techniques used in culturing plasmodia (Chen et al., 2013; Keller & Schoknecht, 1989a; Keller & Schoknecht, 1989b; Liu, Wang & Li, 2010).

Materials & Methods

Morphological studies

The specimens of D. arenosum were obtained from leaves collected in Minqin County by moist chamber culture from 2016 to 2018 (Fig. 1). These specimens (HMJAUM15005–HMJAUM15007) were deposited in the Herbarium Mycology of Jilin Agricultural University (HMJAU). Hoyer’s Mounting Medium was used for mounting myxomycete spores. One hundred myxomycete spores were observed and measured using the ZEISS Axioscope 5 and Axiocam 506 color photographic system. The ornamentation was observed and measured using a Hitachi SU8010 Scanning Electron Microscope (SEM) running at 5 kV to examine the ultrastructure (García-Cunchillos, Estébanez & Lado, 2021; He et al., 2022).

Figure 1 Representative habitats of Minqin County, Gansu province, China.

(A, B) The artificial forest; (C, D) the desert transition zone; (E, F) the farmland; (G, H) the desert.

Water agar culture

Water agar cultures were prepared according to the methods previously described by Haskins & Wrigley de Basanta (2008) and Tran et al. (2012), summarized as follows: 20 g agar (Shanghai Sangon, China) was added to 1,000 mL sterilized water, and sterilized for 20 min at 121 °C. After sterilization, the bottle was transferred to the super clean bench, and the water agar medium was poured into 9 cm glass Petri dishes. Subsequently, spore suspensions were inoculated onto the media. Continuous observation the formation of plasmodia.

Spore germination and single-spore cultivation

Four sporocarps of D. arenosum were placed in 1.5 mL plastic tubes and 500 µL of sterile water was added to each tube. The tubes were incubated in the dark at 25 °C for 2–12 h until spores germinated. Subsequently, they were added to 2.0% water agar medium with pH of 7.0. Spores were released over the water agar medium. After 48 h, transparent phaneroplasmodia appeared on the culture medium. When the plasmodia began to grow on the cultural medium, these plates were immediately placed in a dark incubator at 22–25 °C. After seven days, we transferred the Petri dish to natural light for incubation to stimulate the formation of sporocarps.

DNA extraction and PCR amplification

The DNA was extracted from three to five sporocarps, including the type specimen (HMJAUM15007!), using the DNeasy Plant Minikit (QIAGEN, Shanghai, China), following the manufacturer’s instructions. The small subunit ribosomal DNA (SSU rDNA) sequence fragment was amplified via PCR amplification, with each reaction mixture containing 2.5 µL 10 × PCR Buffer (Mg2+), 2.5 µL dNTP (2.5 mM/mL), 1 µL each primer, 0.25 µL rTaq (5 U/µL), 0.5 µL DNA template (10 ng/µL) and double-distilled water to a final volume of 25 mL. The primers were Phf1b-A (AAAACTCACCAGGTCCAGAT) and JKr-2 (AGGGCAGGGACGCATTC) (Wrigley de Basanta et al., 2015). The reactions were performed with the following program: initial denaturation at 94 °C for 3 min, 30 cycles of 94 °C for 30 s, 58 °C for 30 s, 72 °C for 2.5 min, followed by 72 °C for 10 min. PCR products were sent to Sangon Biotech Co., Ltd. (Shanghai, China) for sequencing to be directly sequenced using the ABI 3730xl DNA analyzer.

Phylogenetic analyses

The newly generated sequences obtained in this study have been deposited in GenBank (Table 1). For the datasets, the alignment was generated using the “L–INS–i” strategy of MAFFT v.7.017 (Katoh & Standley, 2013). Before performing phylogenetic analyses, start and end ambiguous sites were removed, and gaps were manually adjusted to optimize the alignment by BioEdit v7.1.3 (Hall, 1999). Maximum parsimony (MP) methods were employed using PAUP 4.0 to generate the trees (Wilgenbusch & Swofford, 2003). Bootstrap analysis was performed with 1000 replications to evaluate the topological confidence of MP trees. Bayesian inference (BI) phylogeny analyses were conducted using Markov chain Monte Carlo (MCMC) with Mrbayes 3.2.6 (Ronquist & Huelsenbeck, 2003).

Table 1 Taxa information and GenBank accession numbers of the sequences used in this study.

Taxon	GenBank accession	DNA region	Voucher ID	Length (bp)	
Didymium bahiense	AB259387	18S	TNS-M-Y-4944	428	
Didymium clavus	AB259389	18S	JM-4507	423	
Didymium clavus	AB259390	18S	AK-04172	422	
Didymium clavus	AB259391	18S	AK-04296	423	
Didymium clavus	AB259392	18S	TNS-M-Y-17152	422	
Didymium crustaceum	AB259395	18S	TNS-M-Y-17612	425	
Didymium crustaceum	AB259396	18S	YY-26183	425	
Didymium crustaceum	MW404616*	18S	20180821050	429	
Didymium crustaceum	MW404618*	18S	20181007025	429	
Didymium crustaceum	MW404630*	18S	20160923065	431	
Didymium crustaceum	MW404631*	18S	20160704011	428	
Didymium dubium	AB259399	18S	TNS-M-Y-17046	427	
Didymium dubium	AM231294	18S	K7	1932	
Didymium dubium	AM231295	18S	K15	1911	
Didymium arenosum	MW413352 *	18S	HMJAUM15005	419	
Didymium arenosum	MN720571 *	18S	HMJAUM15006	432	
Didymium floccoides	AB259402	18S	AK-04032	428	
Didymium floccoides	AB259403	18S	AK-04046	429	
Didymium floccosum	AB259405	18S	TNS-M-Y-16882	434	
Didymium floccosum	AB259406	18S	JM-3011	434	
Didymium iridis	AB259407	18S	JM-S-08	430	
Didymium iridis	AB259408	18S	JM-643	434	
Didymium laccatipes	AB259410	18S	AK-F028	428	
Didymium marineri	AB259413	18S	TNS-M-Y-15365	426	
Didymium megalosporum	AB259414	18S	JM-S-06	426	
Didymium megalosporum	AB259415	18S	JM-4509	426	
Didymium nigripes	AB435333	18S	AK-05137	433	
Didymium nigripes	AB435334	18S	AK-06080	425	
Didymium nigripes	AB435335	18S	AK-06100	433	
Didymium nigripes	AB435336	18S	AK-06110	433	
Didymium nigripes	MW404632*	18S	HMJAUM15003	438	
Didymium panniforme	AB259428	18S	TNS-M-Y-16880	525	
Didymium squamulosum	AB435337	18S	AK-06063	433	
Didymium squamulosum	AB435338	18S	AK-06085	435	
Didymium squamulosum	AB435339	18S	AK-06119	435	
Physarum roseum	HE614605	18S	C1	1820	
Notes.

Newly generated sequences in this study are in bold.

* Sequence provided in the study.

Results

Taxonomy

Didymium arenosum S.W. Wei, Q. Wang & Y. Li, sp. nov. (Figs. 2A–2I)	

MycoBank: MB849855

Etymology: “arenosum” refers to sandy or full of sand, it is derived from the Latin “arena”, which means sandy place.

Holotype: CHINA, Gansu Province, Minqin County, Harvest Township 103°35′E38°53′N, alt. 1317 m, leaves obtained in the moist chamber culture, pH 7.3, 17 Sept. 2016, Shu-Wei Wei, (Holotype HMJAUM15007!, 18S = OR500607).

Figure 2 Didymium arenosum. (HMJAUM15005).

(A) Sporocarps; (B–E) spores by LM; (F) spore by SEM; (G) stellate calcareous crystals by SEM; (H, I) capillitium by SEM. Scale bars: (A) = 2.5 mm; (B–E) = 5 µm; (F) = 5 µm; (G) = 10 µm; (H) = 5 µm; (I) = 10 µm.

Sporophores sporocarpic, occasionally fused to form short plasmodiocarps, 0.25–0.5 mm high, dispersed or grouped, spherical or flattened, or umbilicate above, greenish-yellow, pale yellow, or yellowish-brown, composed of stellate calcareous crystals, blackish when the crystals are sparse or absent. Sporocarps are sessile or have a short stalk, scattered, globose to subglobose, pulvinate, and plasmodiocarp elliptical or often curved. The centerline is slightly sunken and grows on a colorless and invisible substrate; sessile sporocarps 0.5–0.9 mm diam, whereas plasmodiocarps with a width of 0.3–2.3 mm, and a thickness of 0.3–0.5 mm. Columella flat, dark yellow, or brown. Hypothallus insconspicuous, membranaceous, and individualized to each sporophore. Capillitium filiform, branched, tenuous, and with few cross connections, colorless to light brown, threads 0.5–2.0 µm diam, enlargements 2.7–3.9 µm diam, smooth but with a granular surface by SEM. Spores free, black in mass, dark brown to brown by LM, globose to subglobose, (7.4)7.8 × 9.8(10.8)–(7.7) 8.3 × 10.6(11.2) µm, densely warted by LM, with evenly distributed bacula by SEM, and the spore ornamentation are small warts, although some rounded to irregular, or forms short lines.

Habitats: leaves of Betula platyphylla.

Additional specimens were examined. CHINA, Gansu Province: Minqin County, Harvest Township, 103°35′E, 38°53′N, alt. 1317 m, 13 Sept 2016, Shu-Wei Wei (HMJAUM15005); Suwu Township, 103°6′E, 38°7′N, alt. 1335 m, 17 Sept. 2016, Shu-Wei Wei (HMJAUM15006).

The life cycle

Spore germination occurred via the split method creating a V-shaped opening in sterile water (Figs. 3A–3C). After 3 h of suspension culture, the myxamoeba and swarm cells (Fig. 3D) were released from the spore. The swarm cells, which have two flagella at the anterior end, moved rapidly, while shorter projections were often attached to the side and not easily observed. Under strong light, the spores released their internal contents, gradually growing in size (Fig. 3B) before disappearing (Fig. 3C).

Figure 3 The life cycle of Didymium arenosum.

(A–C) Spores with V-shaped split; (D) swarm cell; (E) young plasmodia; (F) young plasmodia radiate forward; (G) mature plasmodia; (H) young sporangium; (I) mature sporangium. Scale bars: (A, B) = 10 µm; (C) = 8 µm; (D) = 10 µm; (E) = 0.1 cm; (F) = 0.3 cm; (G) = 0.5 cm; (H, I) = 0.5 mm.

Initial plasmodia were observed around 7–15 days after the spore suspension of the swarm cells was transferred to the water agar medium (Fig. 3E). Meanwhile, after 2–4 days of cultivation, the initial plasmodia radiated forward, were colorless, lacked an obvious netted form, and had few branches (Fig. 3F). The thickened reticular mature plasmodium mass at the leading edge could be observed after shading and additional oat feeding after 3–6 days (Fig. 3G). The plasmodia matured, transitioning from milky yellow to brownish-yellow, with a sizeable front sector area and many branches, and distributed on the water agar surface as a network.

The mature plasmodia were cultured at 20–22 °C and exposed to natural light. Within 24–48 h, the mesh-shaped plasmodium began to form sporocarps. The formation of sporocarps varied in time, with some developed within three days, while most remained in the vegetative growth stage or formed scleotia during this period. Incomplete secondary germination without light was possible. With exposure to natural light, the plasmodia masses gradually coalesced, forming more robust veins. The sporangium changed color from milky white to light yellow before gradually blackening (Fig. 3H). When the environment was appropriately dry, yellowish-brown or yellow-green calcareous crystals formed on the surface of the sporocarps (Fig. 3I).

Phylogenetic analyses

In this study, we obtained the SSU rDNA sequences of D. arenosum, D. crustaceum Fr., and D. nigripes (Link) Fr. A total of 36 sequences with 403 positions were analyzed phylogenetically, with a myxomycete from Physarum roseum Berk. & Broome as an outgroup. The GTR+I+G nucleotide substitution model was found to be the best fit. The MP and BI trees had similar topologies, with only minor differences in a few nodes with low MP/BI support. Our phylogenetic analyses revealed that the new species D. arenosum forms an independent branch cluster closely related to D. panniforme J. Matsumoto (Fig. 4).

Figure 4 Bayesian inference (BI) and maximum parsimony (MP) phylogenetic analysis based on the SSU rDNA sequences of Didymium species.

Branches are labeled with Bayesian posterior probabilities greater than 0.95 and maximum parsimony bootstrap support more significant than 70%. The newly generated sequences are bolded, and the new species are indicated in a shadow of gray.

Discussion

Based on the morphological and phylogenetic analyses, this study discovered a new species in the arid region, and completed its life cycle on water agar culture. Morphologically, the sporangium of D. arenosum was smaller than D. ochroideum G. Lister (Lister, 1931), whereas D. arenosum was greenish-yellow, pale yellow, or yellowish-brown, while D. ochroideum was light orange-brown. In macro-morphology, D. arenosum was similar to D. obducens P. Karst, but the color of peridium was different, and D. obducens was pale brownish ochraceous. In micro-morphology, the capillitium of D. obducens was light brown, and the tips were usually darker. Furthermore, D. obducens can be distinguished by the larger spores (12–14 µm). Unlike D. arenosum, the surface of D. panniforme (Mastsumoto & Deguchi, 1999) was covered with orange calcium carbonate crystals, similar to the new species. There were many ridges on the surface of spores, most of that were connected by an incomplete network. In contrast, Didymium arenosum has irregular spines and warts which can be combined into short lines. The sporangium of D. arenosum and D. tussilaginis (Berk. & Broome) Massee are morphologically similar, occasionally fused to form plasmodiocarps, which are difficult to distinguish with the naked eye. However, the inconspicuous columella of D. tussilaginis and the larger spores (12–13 µm) help distinguish it from the new species (Baba et al., 2021). Within the myxomycetes, phylogenetic research is still at a relatively early stage, and our phylogenetic analyses suggested that D. arenosum has close affinities with D. panniforme (Fig. 4), consistent with the morphological study (Leontyev & Schnittler, 2022).

Previous studies have shown that the myxomycetes cultivated in the moist chamber could be found in three months (Wrigley de Basanta et al., 2017). However, it only needs half a month to form the sporangium in this study. The likely explanation is that the species mainly grows in the arid regions. Therefore, the microhabitat changed suddenly in the presence of favorable growing conditions in the moist chamber, resulting in the growth rate of the species significantly accelerated.

In the moist chamber cultures of D. arenosum, the substrate pH ranged from 6.4 to 7.6, with an average pH of 7.0. This range is similar to the circumneutral pH of the agar medium on which it completed its life cycle from spore to spore. Several studies have demonstrated that substrate pH affects the growth of moist chamber cultures (Wrigley de Basanta, 2000; Wrigley de Basanta, 2004). However, some species have adapted to basic media in arid regions. For example, Didymium wildpretii Mosquera, Estrada, Beltrán-Tej., D. Wrigley & Lado grows on substrata with pH values of 7.5–10, and Licea succulenticola Mosquera, Lado, Estrada & Beltrán-Tej also grows on a substrate with a higher pH (Lado et al., 2007; Mosquera et al., 2002). The differences in pH selection might reflect the different organisms during a succession of substrate decay rather than direct effects on the myxomycetes. Both bacteria and yeasts have been shown to have high levels of specialization in the host plant (Wrigley de Basanta, Lado & Estrada-Torres, 2008).

During the life cycle of D. arenosum, germination occurred through a V-shaped split within 3 to 72 h, releasing swarm cells with one long and one short flagellum that swam in the sterilized water. This germination mode is common among many species of Didymium (Gao et al., 2017; Ishibashi et al., 2001; Lado et al., 2007). Similarly, the life cycle of D. arenosum and D. xanthopus (Ditmar) Fr. required light treatment (Gray, 1938). Without light, the plasmodia of D. arenosum would gradually dissolve or form sclerotia in the medium. The life cycle of this species proceeds in a closed system with constant humidity and temperature, so the sclerotia formation was not a response to drying, it might be due to self-protection mechanism. Therefore, the conditions for sporangium formation in D. arenosum depend on external conditions and the abundance of nutrients. Overall, our findings suggest that the life cycle of D. arenosum is similar to that of other Didymium species, and the formation of sporangium requires light. Additionally, the formation of sclerotia in agar culture may be influenced by nutrient availability rather than dampness degree. Further research is needed to elucidate the mechanisms underlying these processes in D. arenosum and other related species.

The germination of D. arenosum, within 72 h, which is similar to another species D. infundibuliforme Wrigley de Basanta et al. (2009). Didymium squamulosum has a life cycle of only 12 days (Zhu et al., 2019), likely influenced by its extensive distribution and need to occupy sufficient ecological niches in the natural ecosystem. Didymium infundibuliforme completed its life cycle in about 50 days (Wrigley de Basanta et al., 2009), while D. umbilicatum D. Wrigley, Lado & Estrada from arid regions of Mexico, takes at least 51 days when cultured (Wrigley de Basanta et al., 2009), and D. wildpretii takes about 28–56 days (Lado et al., 2007). Blackwell & Gilbertson (1980) suggested that a shorter life cycle would be more adaptive in extreme desert conditions. However, the ability and propensity of these species to rapidly generate sclerotia could serve as a more reliable survival mechanism against sudden environmental changes, and these dormant periods would prolong its likelihood of survival.

Conclusions

Through morphological studies, life cycle, and phylogenetic analyses, this study has provided a comprehensive and systematic examination of D. arenosum. The new species demonstrated adaptations to arid conditions, as evidenced by its life cycle and pH levels. The identification of D. arenosum contributes to the expanding knowledge regarding myxomycete diversity in arid regions, where these organisms were previously perceived as rare in desert environments. The rapid formation of swarm cells and sclerotia in D. arenosum likely constitutes a crucial adaptation for coping with environmental fluctuations. Overall, this study not only deepens our understanding of the life cycle and survival strategies of these organisms but also emphasizes the importance of ongoing exploration in arid regions to unveil the diversity and distinctive adaptations of various protist groups, such as myxomycetes.

Supplemental Information

Supplemental Information 1 Sequences

Click here for additional data file.

We sincerely thank Chaofeng Yuan (Engineering Research Center of Edible and Medicinal Fungi, Ministry of Education, Jilin Agricultural University, China) and Wenhe Liu (Engineering Research Center of Edible and Medicinal Fungi, Ministry of Education, Jilin Agricultural University, China) for their help in collecting field specimens. We special thank Professor Harold W. Keller (Botanical Research Institute of Texas, USA) for the valuable comments.

Additional Information and Declarations

Competing Interests

Author Contributions

DNA Deposition

Data Availability

New Species Registration

The authors declare there are no competing interests.

Shuwei Wei performed the experiments, analyzed the data, prepared figures and/or tables, authored or reviewed drafts of the article, and approved the final draft.

Shu Li analyzed the data, prepared figures and/or tables, and approved the final draft.

Pu Liu analyzed the data, prepared figures and/or tables, and approved the final draft.

Bao Qi performed the experiments, authored or reviewed drafts of the article, and approved the final draft.

Qi Wang conceived and designed the experiments, authored or reviewed drafts of the article, and approved the final draft.

Yu Li conceived and designed the experiments, authored or reviewed drafts of the article, and approved the final draft.

The following information was supplied regarding the deposition of DNA sequences:

The new species sequences are available at GenBank: MW413352, MN720571.

The Didymium crustaceum are available at GenBank: MW404616, MW404618, MW404630, MW404631.

The Didymium nigripes are available via GenBank: MW404632.

The following information was supplied regarding data availability:

The raw sequences are available in the Supplemental File.

The following information was supplied regarding the registration of a newly described species:

Mycobank: MB849855

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
