# Peer review of "Didymium arenosum, a myxomycete new to science from the confluence of deserts in northwestern China"

_PeerJ, doi:10.7717/peerj.16725_

## Round 0.1 · original submission · Major Revisions

Authors should address all the queries raised by the reviewers, especially Reviewer 1.

·

Basic reporting

.

Experimental design

.

Validity of the findings

.

Additional comments

General comments authors should consider.

Title: I suggest word changes: Didymium flavovirens, a new myxomycete in China. New what? flavovirens is not the best specific epithet because color is extremely variable in the myxomycetes. Please consider another name based on structural characters. …a myxomycete new to science is better wording and China is such a big country you might consider adding the Alashan desert from northwestern China. A better title reads: Didymium ???, a myxomycete new to science from the Alashan desert from northwestern China.

Abstract: last sentence, The spore to spore agar culture of Didymium sp. requires 23 days, and this study provides a detailed description of its life cycle.

1. Classification, Life Cycles, and Spore to Spore Agar Culture. The two moist important points are the classification of Myxomycetes and the life cycle spore to spore culture on agar and structure of fruiting bodies and degree of calcification of the peridium, and overall size dimensions of fruiting bodies. These characters are highly variable on agar culture. Without field collections of this Didymium species to compare species descriptions, accuracy of descriptive structural characters may not be reliable. To remedy this situation, I strongly urge the authors consult, read, and cite two chapters in this book on slime molds published in 2021:
Keller, H.W., S.E. Everhart, and C.M. Kilgore. 2021. Myxomycetes: Basic Biology, Life Cycles, Genetics and Reproduction. In C. Rojas and Stephenson, S. (eds) “Myxomycetes: Biology, Systematics, Biogeography and Ecology”, Chapter 1, 1–45. SECOND EDITION, Elsevier An Imprint of Academic Press, Atlanta, GA. Please read the entire chapter but especially pages 37 to 39. Also, consult:
Keller, H.W. and J D. Schoknecht. 1989. Spore-to-spore culture of Physarum spinisporum and its transfer to Badhamia. Mycologia. 81: 631–636.
Keller, H.W. and J.D. Schoknecht. 1989. Spore-to-spore cultivation of a new wrinkled-reticulate-spored Badhamia. Mycologia. 81: 783–789.
Chapter 3. The phylogeny and phylogenetically based classification of myxomycetes. In C. Rojas and Stephenson, S. (eds) “Myxomycetes: Biology, Systematics, Biogeography and Ecology”, Chapter 3, 97–124. SECOND EDITION, Elsevier An Imprint of Academic Press, Atlanta, GA.
Classification of Myxomycetes has changed dramatically the past several years, therefore, the authors may wish to update their assessment of their classification with more current references that I have provided here.

The Abstract: The combination of morphological characters do not distinguish this species from other species of Didymium except for color which can be extremely variable in the Myxomycetes. The crystalline calcium carbonate crystal is an important key character that separates the genus Didymium from other calcareous myxomycete genera. How did you demonstrate the calcium carbonate composition (HCL acid or clear lactophenol) with the release of bubbles as the acidification dissolves the crystals? This test should have been done. There was no micrograph, either light microscope or scanning microscope, of the crystals which is needed to document the complete morphospecies description? The spore ornamentation is not distinctive and many other species of myxomycetes have similar spore markings. The spore ornamentation is not unique.

2. Conclusions topical sections? Editorial criteria include this section with this description: conclusions are well stated, linked to original research questions & limited to supporting results. Please add this section to comply with review directives.

3. Introduction: Lines 58-60 fails to include the many publications of Lado and others based on myxomycete collections from desert and arid regions of South America where the species diversity of myxomycetes, especially species new to science, should be referenced. Here is one example but there are others that should be included. When these areas are included species diversity in arid and desert regions of the world are actually surprisingly fairly high. Lado, C., Wrigley de Basanta, D., Estrada-Torres, A. & Stephenson, S.L. 2016. Myxomycete diversity in the coastal desert of Peru with emphasis on the lomas formations. Anales Jard. Bot. Madrid 73(1): e032 This is just one example but there are many others that could be cited as references. This paper would be much more valuable if this region of the world would be included for myxomycete species diversity. Are there any other arid or desert areas of China that are unexplored that need additional myxomycete collections? China is a big country and noting additional arid regions would add more value to the paper. This part of the paper should re revised to include additional commentary and references. Figures 1-4 are nice habit additions that provide landscapes of northwestern China. The authors should be congratulated for including these photographic images.

4. Species description lines 143-153. This used to be a diagnosis in Latin but now according to the Botanical Code of Nomenclature it is in English or Spanish. However, there are some problems with this species description. Line 148 The surface of the sporangia change to The sporangial surface…You can shorten this some more by using semicolons and drop the. Line 149 Capillitium is meticulous does not make sense. The capillitial system could be described as a branching and anastomosing system of threads but nowhere is this described. Attachments are important either to the peridium at the top, sides, or bottom? This is not part of the species description. These capillitial threads should lack calcareous nodules but nowhere is this mentioned. Enlargements add an s. Lines 151 and 152. Spores black in mass not piled up. The spore ornamentation IS NOT UNIQUE but description details are not provided. This species description is incomplete and lacks details to differentiate this species from other species of Didymium. Typically spore measurements are based on at least 100 spores counted with an ocular micrometer. This lack of spore detail results in describing this new to science species as being problematic.

5. The life cycle Lines 160-183 is an added plus and nicely done and represents one of the strengths of this paper. The author’s merit special recognition for their efforts to culture this species.

6. There are some serious shortcomings in the Figures that are mislabeled with numbers. Figure 1 should probably be 1a, 2b, 3c, 4d. Figure 2, 5 sporocarps; 6 , authors 2 are spores showing spherical shape but a substandard photomicrograph failing to show surface ornamentation; 7. Author’s 3 Spore seen with SEM 8. Author’s 4. What is the difference in these two SEMs? Spore surface ornamentation not described accurately. Spore tips? rounded or sharply point (warted or spinulose) SEMs magnification not high enough to tell?; ornamentation forms short lines but the literature search was incomplete to make the statement that the spores were unique?; Capillitial thread 9 Author’s 5 Thread simple branch, surface appears smooth; 10 Author’s 6, capillitial threads mostly unbranched slightly roughened. There was no description

7. What kind of mounting medium was used to make the spore measurements? How many spores were measured? These are basic elements of the Materials and Methods of any taxonomic paper describing myxomycete fruiting bodies. Lines 93-100. Martin and Alexopoulos 1969 is cited as a literature source for SEM and that publication did not use an SEM. In fact spore measurements were made using a 430X compound microscope with high power and dry objective lens not oil immersion lens.

8. The life cycle. The split v-shaped method of spore germination is typical for species in the Physarales; as compared to the pore method, for example, in the Liceales. However, what percentage of the spores germinated? Did every spore germinate? You did not indicate the ingredients for the water agar medium? You say .75% water agar medium but usually it is 2%. That low a percentage would result in soft mushy agar? Are you sure this is correct because I always used at least 2% water agar. What was the source of the agar? How many agar plates were prepared and how many produced myxomycete fruiting bodies? This is important when recording the number of fruiting bodies observed and the basis for measurements. Usually many of the prepared plates result in different stages of development but not all go from spore to spore and produce fruiting bodies.

9. This taxon was compared to Didymium saturnus but the peridial crystals are not similar and neither are the spores. Didymium saturnus spores were unique because the elliptical spore had a conspicuous ring around the equator much like the rings of the plant Saturn hence the name. I doubt that D. saturnus should be compared morphologically here. I know something about this taxon because I described the species new to science and cultured it from spore to spore.

10. I need to comment about the DNA and phylogram. Inclusion of this molecular evidence is something special because it provides newer techniques and evidence that whatever this life form is it is different. However, the author’s have not made an acceptable case for a new species based on the incompleteness of the species description and lack of field collections.

11. Figure 3. I like these developmental stages and the photographs demonstrate nicely all of the stages shown. 11, 12, and 13 show a germinating spore with ornamentation because microcysts are smooth and lack ornamentation. It would help if the authors could put a day-old age for the plasmodium (see 15, 16, and 17. Fig 18 is probably not a sclerotium but it is difficult to say with confidence what this represents. 19 is an immature sporangium and 20 is a circular mature sporangium but the photograph does not show the typical color of the crystals. No photograph was provided of the crystals which is probably the basis for the surface coloration of the sporangium.

12. Except for a few places the English is quite good and acceptable. Basic Reporting, Validity of Findings, Experimental Design, and Conclusions. Assessment of acceptability of this paper at this time based on the species description does not meet the criteria and standards for a species new to science, and therefore, needs revision and resubmission. I strongly urge the authors to visit the arid areas after rainy periods to see if they can find this taxon as field collections and then compare them to cultured spore to spore collections. Myxomycete fruiting bodies developed on agar culture are important sources of additional information but lack of a complete and accurate morphological species description does not warrant a species new to science at this time.

Reviewer 2 ·

Basic reporting

Line 66 Scientific name should be written in italics. Same in line number 103.
Line 95 Specimen deposition number/catalogue number should be mentioned.
Line 104 Change “hour/hours” to “h”. Follow the style throughout the manuscript.
Line 113 Check the type specimen “ID HMJAUM15007!”. I could not find it in the NCBI database.
Line 115 Change “Mg2+ plus” to “Mg2+”
Line 119 Change “3min” to “3 min”. Follow the style throughout the manuscript.
Line 220 Change "Ph" to "pH". Follow the style throughout the manuscript.

Experimental design

No comment

Validity of the findings

All the GenBank accession numbers provided in Table 1 are accessible at NCBI database. However, some of the voucher IDs are not provided. Can the authors provide all the voucher ID in Table 1?

Additional comments

The authors have done a great job in providing detailed descriptions of morphology and life cycle of Didymium flavovirens.
I request the authors to provide a higher resolution or higher quality picture of Figure 20.

---

## Round 0.2 · Minor Revisions

This manuscript is much improved but the rewritten text is in substandard English which the reviewer tried to improve. This manuscript is NOT ACCEPTABLE in its present form and the reviwer requests a newly submitted manuscript with Figures and Captions. Please find the Annotated PDF in the attachment and address all the queries/suggestions given by the reviewer 1.

**Language Note:** The Academic Editor has identified that the English language must be improved. PeerJ can provide language editing services - please contact us at copyediting@peerj.com for pricing (be sure to provide your manuscript number and title). Alternatively, you should make your own arrangements to improve the language quality and provide details in your response letter. – PeerJ Staff

·

Basic reporting

.

Experimental design

.

Validity of the findings

.

Additional comments

See attachment

Reviewer 2 ·

Basic reporting

The authors have addressed the comments made by Reviewer 2 and made corrections accordingly.

Experimental design

The authors have addressed the comments made by Reviewer 2 and made corrections accordingly.

Validity of the findings

The authors have addressed the comments made by Reviewer 2 and made corrections accordingly.

---

## Round 0.3 · accepted · Accept

Revisions made in response to the reviewers' suggestions were found to be satisfactory, The authors have addressed all of the reviewers' comments and I am happy with the current version. The manuscript is ready for publication.